# Glucosylceramide Changes Bacterial Metabolism and Increases Gram-Positive Bacteria through Tolerance to Secondary Bile Acids In Vitro [note 1]

**DOI:** 10.3390/ijms23105300

**Published:** 2022-05-10

**Authors:** Huanghuang Dai, Akira Otsuka, Kurumi Tanabe, Teruyoshi Yanagita, Jiro Nakayama, Hiroshi Kitagaki

**Affiliations:** 1The United Graduate School of Agricultural Sciences, Kagoshima University, Korimoto 1-21-24, Kagoshima 890-0065, Japan; 21975002@edu.cc.saga-u.ac.jp; 2Faculty of Agriculture, Saga University, Honjo-cho, 1, Saga 840-8502, Japan; yanagitt@gmail.com; 3Graduate School of Health Sciences, Saga University, Honjo-cho, 1, Saga 840-8502, Japan; 21626006@edu.cc.saga-u.ac.jp (A.O.); 22626010@edu.cc.saga-u.ac.jp (K.T.); 4Department of Health and Nutrition Sciences, Nishikyushu University, Ozaki, 4490-9, Saga 842-8585, Japan; 5Division of Systems Bioengineering, Department of Bioscience and Biotechnology, Faculty of Agriculture, Graduate School, Kyushu University, Motooka 744, Nishi-ku, Fukuoka 819-0395, Japan; nakayama@agr.kyushu-u.ac.jp

**Keywords:** glucosylceramide, prebiotics, intestinal microbes, bile acids, *Blautia coccoides*

## Abstract

Glucosylceramide is present in many foods, such as crops and fermented foods. Most glucosylceramides are not degraded or absorbed in the small intestine and pass through the large intestine. Glucosylceramide exerts versatile effects on colon tumorigenesis, skin moisture, cholesterol metabolism and improvement of intestinal microbes in vivo. However, the mechanism of action has not yet been fully elucidated. To gain insight into the effect of glucosylceramide on intestinal microbes, glucosylceramide was anaerobically incubated with the dominant intestinal microbe, *Blautia coccoides*, and model intestinal microbes. The metabolites of the cultured broth supplemented with glucosylceramide were significantly different from those of broth not treated with glucosylceramide. The number of Gram-positive bacteria was significantly increased upon the addition of glucosylceramide compared to that in the control. Glucosylceramide endows intestinal microbes with tolerance to secondary bile acid. These results first demonstrated that glucosylceramide plays a role in the modification of intestinal microbes.

## 1. Introduction

The importance of diet in health has long been recognized worldwide. The Chinese ancient document, the Shurai, written in 800 BC, describes medical doctors who specialized in foods, and the ancient Greek medical doctor, Hippocrates (460–370 BC), described the importance of food in medicine. After the scientific approach was established, the relationship between health and intestinal microbes was studied by Mechnikov, a doctor who, in 1907, proposed that intestinal microbes lower the pH of the intestinal environment and prevent intestinal decay and aging [1]. These microbes were later classified as probiotics. In 1998, Gibson proposed that some substances promote the growth of preferable intestinal bacteria and termed them prebiotics [2]. Fructooligosaccharides [3], galactooligosaccharides [4], beta-glucan [5], polyunsaturated fatty acids [6], catechin [7,8,9], anthocyanins [10], pectin [11], alginate [3], and enzymes [12,13,14] have been proposed as prebiotics.

Sphingolipids are lipids composed of a sphingoid base, an amide-linked fatty acid moiety, and other sensory bases. Since the discovery by Thudichum in 1874, sphingolipids have attracted attention because of their relevance to several neural disorders, such as Gaucher disease, Niemann–Pick, Fabry, Tay–Sachs, Krabbe, and Sandohoff diseases in the 1930s and their chemical structures were determined [15]. In the late 1980s, sphingolipids emerged as anti-cancer agents. For example, the degraded products of sphingolipids, including sphingosine and lysosphingolipids, inhibit protein kinase and have anti-tumor effects [16] and ceramide formed through the subcellular localization regulates stress responses [17].

Glucosylceramide is one of the sphingolipids composed of a sugar moiety and ceramide. In glucosylceramides from *koji*, *Aspergillus oryzae* [18], *A. luchuensis* [19,20,21], and *Cryptococcus neoformans* [22], a simple β-glucose is covalently attached to the 1-hydroxyl base of ceramide. Glucosylceramide has been detected in several species, including crop plants and fungi, as well as fermented foods [23,24,25]. Glucosylceramide prevents aberrant crypt foci [26,27,28,29,30], improves lipid metabolism [31] and skin barrier function [32,33]. To explain the underlying mechanism, glucosylceramide was proposed to prevent the production of inflammatory cytokines in the small intestine [34]. Glucosylceramide-derived sphingoid bases induce apoptosis [35,36], whereas feeding with glucosylceramide increases intestinal *Blautia coccoides, Bacteroides sartorii*, and *Hathewaya histolytica* levels in vivo [23]. Most food components alter their chemical forms in the small intestine through the action of enzymes. However, the majority of the chemical form of glucosylceramide is not altered by enzymes in the small intestine [23]. Indeed, the enzymatic degradation and absorption of glucosylceramide are limited [37,38]. Several intestinal microbes have been shown to degrade glucosylceramide, however, some glucosylceramide remains in its chemical form throughout the colon [39].

Bile acids are synthesized in a glycine- or taurine-conjugated form in the liver, pooled in the bladder, secreted into the duodenum, and degraded into non-ester forms by intestinal microbes [40]. Bile acids facilitate the saponification and absorption of lipids in the small intestine. However, they also disrupt the membrane structure of intestinal microbes. In particular, when cholic acid is dehydroxylated at the 7α-position by intestinal microbes and becomes secondary cholic acid, namely deoxycholic acid, these products exert strong antimicrobial activity and affect intestinal microbial flora in vivo [41]. Secondary bile acids also induce colon cancer [42] and are associated with nonalcoholic steatohepatitis [43].

To gain insight into the role of glucosylceramide in vivo, glucosylceramide was added to intestinal microbes in vitro and cultured anaerobically, and its effects on intestinal microbes and metabolites were analyzed. We found that glucosylceramide alters the metabolism of and production of lactic acid by *Blautia coccoides*, increases the ratio of Gram-positive bacteria and endows tolerance to deoxycholic acid in certain bacteria such as *Lactobacillus delbrueckii, Streptococcus mutans, Clostridium butyricum, Blautia coccoides* and *Enterococcus faecalis.* These novel findings provide information on the mechanism of action of glucosylceramide on intestinal microbes in vivo.

## 2. Materials and Methods

### 2.1. Strains

The following 15 strains were used as the cell-mock-1, which are bacteria residing with human beings commercially available from National Institute of Technology and Evaluation, Japan (Kisarazu, Japan, https://www.nite.go.jp/data/000107977.pdf, accessed on 5 May 2022). *Bacillus subtilis* (NBRC13719), *Bifidobacterium pseudocatenulatum* (NBRC113353), *Clostridium butyricum* (NBRC13949), *Corynebacterium striatum* (NBRC15291), *Cutibacterium acnes* subsp. *acnes* (NBRC107605), *Lactobacillus delbrueckii* subsp. *delbrueckii* (NBRC3202), *Streptococcus mutans* (NBRC13955), *Staphylococcus epidermidis* (NBRC100911), *Acinetobacter radioresistens* (NBRC102413), *Bacteroides uniformis* (NBRC113350), *Enterocluster clostridioforme* (NBRC113352), *Comamonas terrigena* (NBRC13299), *Escherichia coli* (NBRC3301), *Parabacteroides distasonis* (NBRC113806), and *Pseudomonas putida* (NBRC14164). *Enterococcus faecalis* (NBRC 100481) was commercially purchased from the National Institute of Technology and Evaluation. *Blautia coccoides* (JCM1395) was commercially purchased from the Japan Collection of Microorganisms, RIKEN BRC (Wako, Kyoto, Japan).

### 2.2. Materials

Deoxycholic acid was purchased from Nacalai Tesque, Inc. (Kyoto, Japan) and glucosylceramide from soybeans was purchased from Nagara Science Co., Ltd. (Gifu, Japan).

### 2.3. Bacterial Cultures

*Lactobacillus delbrueckiii* subsp. *lactis* was cultured in MRS broth (FUJIFILM Wako Pure Chemical Industries, Ltd., Osaka, Japan). *Escherichia coli, Bacteroides uniformis*, and *Blautia coccoides* were cultured in GAM broth (Nissui, Tokyo, Japan). *Staphylococcus epidermidis* was cultured in No. 702 broth (FUJIFILM Wako Pure Chemical Industries, Ltd., Osaka, Japan). *Clostridium butyricum* was cultured in CM0149 broth (Kanto Chemical Co. Inc., Tokyo, Japan). *Corynebacterium striatum* was cultured in Trypic Soy broth (Becton Dickinson and company, Sunnyvale, CA, USA). *Streptococcus mutans* was cultured in broth containing Tryptic soy broth of 30 g/L and yeast extract of 3 g/L. Other bacteria were cultured in the yeast casitone fatty acids (YCFA) medium and vitamin solution prepared as previously described [44]. Bacterial growth was monitored at 600 nm using a spectrophotometer (UV-1800, Shimadzu Co., Kyoto, Japan).

A glass vial (10 mL, 45 mm × 22 mm, Agilent Technologies Inc., Santa Clara, CA, USA) was used for anaerobic culture. Nitrogen gas infusion for 5 min was used to purge air. The vial was then locked using a PTFE/butyl cap and autoclaved. All bacteria were grown at 37 °C for 24 h under anaerobic conditions using resarulin (final concentration 1 μg/mL) as an indicator of an anaerobic environment.

### 2.4. Coculture of Glucosylceramide and Microbes

Microbes were inoculated into appropriate media and incubated at 37 °C for total 48 h. Cell-mock-1 was incubated as a mixture of 15 bacteria. Other bacteria were incubated individually. Considering that the deoxycholic acid concentration in the colon is 0.046–0.21 mM [40], the final concentrations of deoxycholic acid in the broths were adjusted to 0–2.0 mM. Deoxycholic acid (2 mM in distilled water) was diluted to the appropriate concentration, according to the experimental conditions. Using a syringe, 100 μL of pre-configured deoxycholic acid (24 h from the start of the culture), 25 μL of the glucosylceramide solution (20 μg/mL dissolved in ethanol or DMSO), vitamin solution (10 μL/mL) and a certain volume of the bacterial culture broth (OD_600_ = 0.1) were added individually to the sterilized broth. The total volume of the liquid in the vial was 5 mL.

### 2.5. Analysis of Metabolites of Cultured Broth Using Gas Chromatography

The cultured broth was freeze-dried, and water-soluble metabolites were extracted and analyzed using gas chromatography with flame ionization detection (GC-FID), as previously described [45].

### 2.6. Analysis of Lactic Acid

The lactic acid concentration in the cultured broth was analyzed using high-performance liquid chromatography (HPLC), as previously described [46].

### 2.7. Next-Generation Sequencing Analysis of Cell-Mock-1 Incubated with Glucosylceramide

Cultured broth (5 mL) was collected and frozen at −27 °C. Next-generation sequencing (NGS) analysis was performed using the Techno Suruga Laboratory (Shimizu, Japan). Briefly, bacterial DNA was extracted using the MORA-EXTRACT kit (Kyokuto Pharmaceutical, Tokyo, Japan) and FastPrep-24 5G (MP Biomedicals, Irvine, CA, USA). Conditions for PCR amplification of 16S rDNA including primers are described by Takahashi et al. [47]. The PCR products were quantified, and deletion of primer sequences and determination of the sequences was performed using the MiSeq system (Illumina, San Diego, CA, USA) and MiSeq Reagent Kit v3 (600 cycles) (Illumina, San Diego, CA, USA). Fastq-join [48] was used to join pair-ends, and the FASTX-Toolkit [49] was used to filter sequences with more than 99% quality values (>20). QIIME ver. 1.8.0 [50] was used for the deletion of chimeric sequences detected by Usearch6.1.544_i86. The RDP MultiClassifier ver. 2.11 was used to search for homologous sequences with a confidence of more than 0.8.

### 2.8. Statistical Analysis

IBM SPSS Statistics (ver. 23.0; IBM Inc., Armonk, NY, USA) and SIMCA-P+, ver. 13.0 (Umetrics, Umeå, Sweden) were used for statistical analyses. Normality of the data was verified using the Shapiro–Wilk test. Significance of differences between the means was verified using a one-sided unpaired Student’s *t*-test. Hierarchical clustering and principal component analysis were used to group the operational taxonomic units (OTUs) of the NGS analyses. Hierarchical clustering, principal component analysis, and partial least squares-discriminant analysis (PLS-DA) were used to analyze the data obtained by GC-mass spectrometry (MS). Hierarchical clustering was performed using MetaboAnalyst ver. 5.0 (https://www.metaboanalyst.ca/home.xhtml, accessed on 5 May 2022) [51]. The OTU counts of the 16 bacterial species were divided by the total OTU count. IC_50_ values were calculated using the ImageJ software. Curve fitting (four-variable logistic curve) with the ImageJ software was used to create approximate dose-response curve equations. The concentration of deoxycholic acid at a 50% reduction in bacterial concentration was considered the semi-inhibitory concentration of the bacteria. Each experiment was repeated three times, and the average value was calculated.

## 3. Results

### 3.1. Glucosylceramide Addition Alters Metabolism of Blautia coccoides

As most glucosylceramide reaches the colon without enzymatic degradation, glucosylceramide was directly added to the in vitro culture of intestinal microbes and analyzed as a model large intestine. The effect of culturing *Blautia coccoides*, a major intestinal microbe involved in the decrease in visceral fat accumulation [52], with purified glucosylceramide was analyzed in vitro. To determine the effect of glucosylceramide, metabolites of *B. coccoides* were incubated with or without the addition of glucosylceramide. Considering that glucosylceramide is contained in crops and fermented foods at a concentration of 0.94–2.2 mg/g [19,24,25,53], the concentration was set at a final concentration of 20 μg/mL, since it can be hypothesized that 25 g of crops or fermented foods reach the large intestine, which has an average capacity of 1065 cc [54] (corresponds to 21 μg/mL in the large intestine if the food contains 10 % water). The cocultured metabolites were analyzed and compared (*n* = 8, Appendix A). Consequently, 32 peaks were identified (Table 1). Based on these data, PLS-DA was performed and corrected using an internal standard approach (Figure 1A), which revealed that the control and glucosylceramide groups were separated (Figure 1C). The permutation test results demonstrated that this was not an overfitting model (Figure 1D). Important components that led to the separation of the two groups, as indicated by a VIP score greater than 1.0, were chosen. Substances, such as citric acid, threonine, lysine, lactic acid, valine, proline, and isoleucine, contributed to the separation of the two groups (Table 2). A one-sided unpaired Student’s *t*-test revealed a substantial increase in citric acid content in the glucosylceramide group compared with that of the control group (*p* < 0.01). To further elucidate the effect of glucosylceramide on the physiology of *B. coccoides*, the short-chain fatty acids produced by *B. coccoides* were analyzed. Lactic acid levels were significantly increased in the broth supplemented with glucosylceramide compared with that of the control broth (Figure 1E). Taken together, these results indicate that glucosylceramide addition alters the metabolism of *B. coccoides*.

### 3.2. Glucosylceramide Addition Increases Gram-Positive Intestinal Microbes

These results clearly indicate that glucosylceramide alters the metabolism of the intestinal microbes in this system. Next, to determine which bacteria were increased in response to glucosylceramide, NGS analysis was performed after incubation of model intestinal microbes (cell-mock-1), consisting of known ratios of bacteria that reside within human beings with glucosylceramide (Appendix A). Cell-mock-1 is a composite of bacteria *Bacillus subtilis*, *Bifidobacterium pseudocatenulatum*, *Clostridium butyricum*, *Corynebacterium striatum*, *Cutibacterium acnes* subsp. *acnes*, *Lactobacillus delbrueckii* subsp. *delbrueckii*, *Streptococcus mutans*, *Staphylococcus epidermidis*, *Acinetobacter radioresistens*, *Bacteroides uniformis*, *Clostridium clostridioforme*, *Comamonas terrigena*, *Escherichia coli*, *Parabacteroides distasonis*, and *Pseudomonas putida*. These bacteria were incubated in YCFA medium as a mixed state anaerobically with or without glucosylceramide. From the incubated culture, genomes were extracted and analyzed using NGS. As a result, relative OTU counts of *Clostridium sensu stricto* and *Escherichia/Shigella* were dominant (Figure 2A), maybe because of the anaerobic condition. These dominant bacteria were not significantly affected by glucosylceramide. In contrast, relative OTU counts of minor Gram-positive bacteria, such as *Lactobacillus delbrueckii* (Figure 2B), *Staphylococcus epidermidis* (Figure 2C), *Streptococcus mutans* (Figure 2D), *Corynebacterium striatum* (Figure 2E), and *Clostridium butyricum* (Figure 2F), were significantly increased in glucosylceramide-treated cultures compared with those in control cultures (*p* < 0.05). In contrast, the Gram-negative bacterium, *Acinetobacter radioresistens* (Figure 2G), decreased in glucosylceramide-treated cultures compared with that in control cultures (*p* < 0.05).

### 3.3. Glucosylceramide Confers Tolerance of Intestinal Microbes to Secondary Bile Acids

Considering that certain intestinal bacteria have a tolerance to bile acids [55], we hypothesized that these bacteria increased growth in the existence of glucosylceramide because they acquired tolerance to deoxycholic acid. Therefore, tolerance of these bacteria towards deoxycholic acid was investigated.

With the addition of glucosylceramide, *L. delbrueckii* showed increased growth at 1.0–2.0 mM deoxycholic acid relative to that in the control (Figure 3A). Glucosylceramide-supplemented *L. delbrueckii* displayed an increased half-maximal inhibitory concentration value compared with that of the control, although the difference was not statistically significant (Table 3).

The tolerance of another bacterium, *S. epidermidis*, to deoxycholic acid was also investigated. There was no significant difference in the growth or half-maximal inhibitory concentration between the glucosylceramide-treated and control *S. epidermidis* (Figure 3B and Table 3). Growths of *S. mutans*, *C. striatum* and *C. butyricum* in the presence of deoxycholic acid were also partially rescued with glucosylceramide (Figure 3C–E). IC_50_ of *S. mutans* was significantly increased (*p* < 0.01, Table 3). The growth of *E. coli* was not inhibited by deoxycholic acid (Figure 3F) and it was difficult to measure the effect of glucosylceramide. The glucosylceramide effect was not observed in *Bacteroides uniformis* (Figure 3G). Furthermore, the tolerances of other major intestinal microbes, including *Blautia coccoides* and *Enterococcus faecalis*, against deoxycholic acid were investigated. The growth of *B. coccoides* was rescued with glucosylceramide at 1 mM deoxycholic acid concentration (Figure 3H). Similarly, the growth of *E. faecalis* was partially rescued with glucosylceramide (Figure 3I). The IC_50_ value for *E. faecalis* was also significantly higher after glucosylceramide addition than that in the control (Table 3). These results suggest that glucosylceramide increased the tolerance of certain intestinal microbes to deoxycholic acid.

## 4. Discussion

Although we have long known that glucosylceramide is present in foods such as crops and fermented foods at 0.94–2.2 mg/g [19,24,25,53], and the majority of the molecule passes through the small intestine and reaches the colon, the mechanism of action of glucosylceramide on intestinal microbes has not yet been reported. In this study, we first report the in vitro prebiotic effects of glucosylceramide. Using metabolome analysis followed by PLS-DA, we demonstrate that adding glucosylceramide to the major intestinal bacterium *B. coccoides* has a significant metabolic effect. Furthermore, the addition of glucosylceramide increased the number of Gram-positive bacteria through tolerance to deoxycholic acid. These results provide the first evidence of a mechanism underlying the action of glucosylceramide on intestinal microbes.

This research has certain limitations and weaknesses. Adding pure components to monocultures does not necessarily reflect what occurs in the complex microbiota of the gut, since the actual gut is filled with various substances and microbes. Moreover, a certain amount of glucosylceramide, although very small [38], might be degraded and absorbed in the small intestine. This research did not consider the absorption of glucosylceramide into the epithelial cells or mucosa in the intestine into account. However, it has a strong aspect as relative to in vivo studies. During the passage through the intestine, glucosylceramide might interact with many bacteria. Some bacteria might attach to the epithelial cells of the intestine, and might not be detected in the feces. On the contrary, in vitro studies generates the direct interaction knowledge between glucosylceramide and bacteria. Therefore, the knowledge obtained through in vitro study should mutually complement in vivo studies to reach a comprehensive understanding of the interaction between substances and intestinal microbes.

Secondary bile acids are formed in the large intestine by intestinal microbes and disrupt the membranes of bacteria, killing them [41]. In the present study, glucosylceramide attenuated these effects. Therefore, glucosylceramide that reaches the large intestine may protect the intestinal microbes from secondary bile acids.

*Blautia* species are one of the most dominant intestinal epithelial barrier-associated bacteria, which belongs to the *Lachnospiraceae* family and *Firmicutes* phylum [56]. *Blautia* species are strictly anaerobic, non-motile bacteria with spherical or oval morphologies. They can assimilate various carbohydrates such as glucose, fructose, lactose, mannose, arabinose and xylose, CO and H_2_/CO_2_, and produces acetic acid, succinic acid, lactic acid and ethanol. It is suggested from gene expression analysis that the tendency of assimilation of carbohydrates of *Blautia* species is lower than that of lactic acid bacteria. There is a negative correlation between the abundance of Blautia and the markers of obesity-related metabolic disorders when fed with maize-derived non-digestible feruloylated oligo- and polysaccharides [57], potato fibers [58] and soy milk [59]. Intestinal microbes of Japanese people contained a high abundance of *Blautia* and *Bifidobacterium* and a low abundance of *Bacteroides* [60,61]. Considering that *Blautia* has a strong taxonomic association with twin inheritance [62], there might be a genetic preference of Blautia towards the intestine of specific genetic background. *Blautia* species are also linked to abnormal Paneth cell counts [63], Crohn’s disease and primary sclerosing cholangitis [64]. *B. coccoides* in the intestine increases with glucosylceramide in an animal model [23], but decreases with pectins and flavanones in humans [65]. *Blautia* species produce useful secondary metabolites such as chrysin, apigenin, desmethyllicaritin, 3′desmethyllarctigenin, bisdemethylcurcmin, demethyldemethoxycurcumin and ceramide in the intestine [54].

Promoted survival of Gram-positive bacteria elucidated in this study suggests the clinical benefits of glucosylceramide. For example, feeding of *L. brueckii subsp. delbrueckii* improves immune health in an animal model [66] and inhibits the growth of *Klebsiella pneumonia* [67]. *C. butyricum* also functions as probiotics by inducing interleukin-10-producing macrophages in inflamed mucosa via the Toll-like receptor 2/myeloid differentiation primary response gene 88 pathway [68] and has many clinical applications [69]. Although pathogenicity is reported for *C. striatum* [70], these events are not reported in the intestine. Since *S. mutans* [71] and *S. epidermidis* [72] form biofilms and are opportunistic pathogens, the health benefits of the modulation of intestinal microbes through the ingestion of glucosylceramide should be considered comprehensively. Still, this study suggests that the ingestion of glucosylceramide, on the whole, might promote the survival of beneficial intestinal microbes, which is a target of future studies.

This study did not elucidate how or why glucosylceramide increased the ratio of Gram-positive bacteria. Since Gram-positive bacteria have a thick peptidoglycan layer, they may be better suited for the attachment of glucosylceramide and its incorporation into their structures as compared with that of Gram-negative bacteria. Consistent with this hypothesis, *Lacticaseibacillus casei* was reported to bind glucosylceramide [73]. In addition, several other lactic acid bacteria have been shown to bind sphingolipids containing carbohydrate moieties, including *Lactobacillus johnsonii* to asialoganglioside [74] and *Propionibacterium freudenreichii* to lactosylceramide [75]. Therefore, it can be hypothesized that glucosylceramide bound to the surface of Gram-positive bacteria and endowed tolerance to deoxycholic acid. The effect of glucosylceramide on other intestinal bacteria should also be investigated.

This study proposes a potential mechanism for the observed effects in vivo [26,27,28,29,30]. Our results suggest that glucosylceramide protects intestinal microbes, especially Gram-positive bacteria, increases the production of substances such as citric acid, and as a result, might exert various health effects. Consistent with this hypothesis, citric acid improves several nutritional profiles in quails [76]. However, this hypothesis requires further investigation.

Our analysis of bacterial tolerance to deoxycholic acid was generally consistent with our results obtained from NGS analysis. However, *S. epidermidis* did not show significant recovery of tolerance to deoxycholic acid with glucosylceramide, which was inconsistent with the NGS analysis. The tolerance analysis may have lacked sufficient sensitivity to detect the difference in *S. epidermidis* tolerance towards deoxycholic acid, which can be elucidated using a more precise analysis tool in future studies.

Several substances have been proposed as potential prebiotics. However, there are no reports of the effects of glucosylceramide on intestinal microbes. Based on our results, glucosylceramide can be hypothesized as a type of prebiotic.

Feeding of fructooligosaccharide and galactooligosaccharide decreased butyrate, increased Actinobacteria, Bifidobacterium, Acidaminococcosu and decreased Salmonella, Coprococcus, Turicbacter, Enterobacter and Phascolactobacterium in vivo [77]. Therefore, it could be hypothesized that glucosylceramide might function differently from oligosaccharides, which is a target of future studies.

Sphingolipid synthesis was shown to improve the survival of intestinal *Bacteroides* [78]. Therefore, it is suggested that exogenous glucosylceramide also improves the survival of these intestinal bacteria, which is a target of future studies.

Glucosylceramide in the luminal side of the intestine, which was investigated in this study, might affect that in the epithelial cells. Glucosylceramide is shown to protect CaCO_2_ cells treated with lipopolysaccharide in an in vitro intestinal tract model [79]. Furthermore, glucosylceramide production maintains colon integrity when challenged with *Bacteroides fragilis* toxin [80]. Together with the results shown in this study that glucosylceramide protects intestinal microbes, glucosylceramide exerts its effects both in the luminal space of the intestine and the epithelial cells.

In conclusion, we demonstrate that the addition of glucosylceramide induces changes in the metabolism of *B. coccoides* and increases the number of Gram-positive bacteria in an intestine-simulating environment through tolerance to deoxycholic acid. These results first indicate the prebiotic-like function of glucosylceramide.

## Figures and Tables

**Figure 1 ijms-23-05300-f001:**
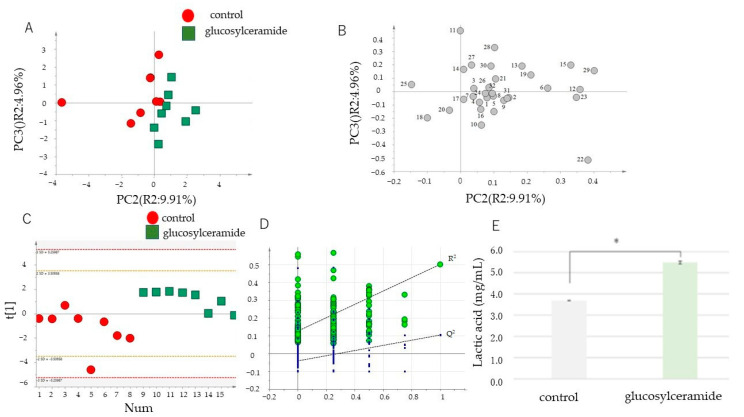
Metabolite analysis of the *Blautia coccoides* culture treated with glucosylceramide. (**A**) Score plot of principal component analysis (PCA) of metabolites of *B. coccoides* treated with or without 20 μg/mL glucosylceramide. (**B**) Loading plot of metabolites of *B. coccoides* treated with or without 20 μg/mL glucosylceramide. The substances corresponding to the numbers are listed in Table 1. (**C**) Partial least squares-discriminant analysis (PLS-DA) of metabolites of *B. coccoides* treated with or without 20 μg/mL glucosylceramide. (**D**) The validated plot of PLS-DA of metabolites of *B. coccoides* treated with or without 20 μg/mL glucosylceramide after a permutation test (*n* = 200). (**E**) Lactic acid concentration (mg/mL) of *B. coccoides* cultures (24 h) treated with or without 20 μg/mL glucosylceramide and 2 mM DCA from the start of the culture in YCFA medium. Statistical difference of the means was verified using one-sided unpaired Student’s *t*-test (*n* = 3, * *p* < 0.05).

**Figure 2 ijms-23-05300-f002:**
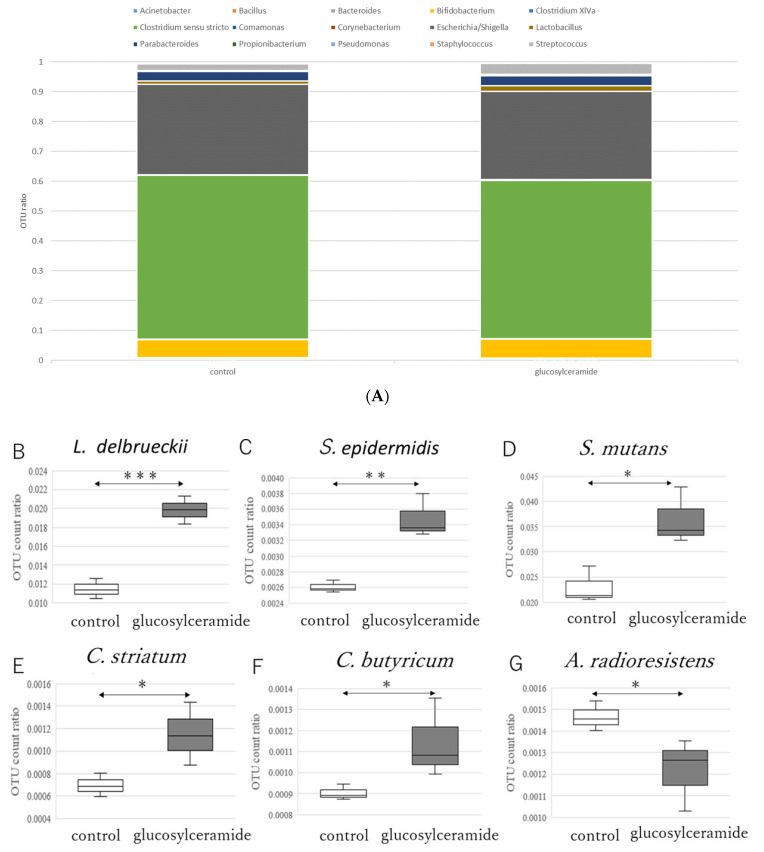
Relative OTU ratios of microbes treated with glucosylceramide. (**A**) Average relative OTU ratios of all microbes treated with glucosylceramide. Relative OTU ratios of (**B**) *L. delbrueckii*, (**C**) *S. epidermidis*, (**D**) *S. mutans*, (**E**) *C. striatum*, (**F**) *C. butyricum* and (**G**) *A. radioresistens* treated with glucosylceramide. Data are expressed as means ± standard errors. Data normality was verified using the Shapiro–Wilk test. Statistical differences in means of data with normalities were analyzed using a one-sided unpaired Student’s *t*-test (*n* = 3; * *p* < 0.05, ** *p* < 0.01, *** *p* < 0.001).

**Figure 3 ijms-23-05300-f003:**
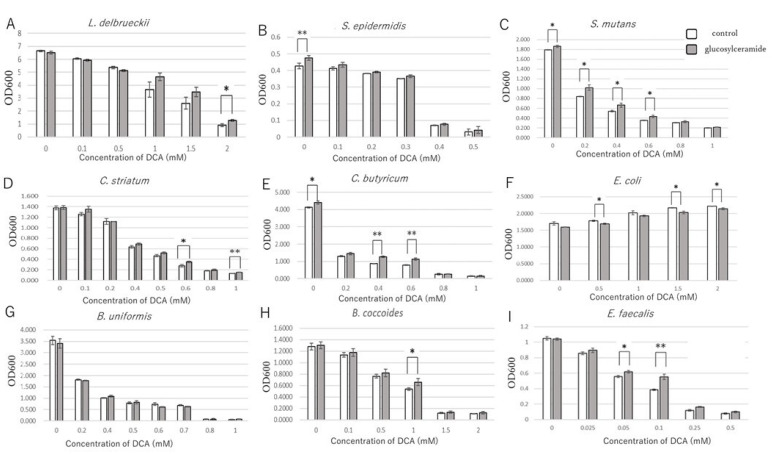
Growth of bacteria supplemented with various concentrations of deoxycholic acid (DCA) and glucosylceramide. OD_600_ values for (**A**) *L. delbrueckii*, (**B**) *S. epidermidis*, (**C**) *S. mutans*, (**D**) *C. striatum*, (**E**) *C. butyricum*, (**F**) *E. coli*, (**G**) *B. uniformis*, (**H**) *B. coccoides* and (**I**) *E. faecalis.* Values are mean ± standard errors of triplicate independent cultures. Statistical differences of the means were evaluated by one-sided unpaired Student’s *t*-test (*n* = 3, * *p* < 0.05, ** *p* < 0.01).

**Table 1 ijms-23-05300-t001:** Substances detected by GC-FID.

No.	Retention Time (min)	Peak Name
1	6.923	Lactic acid
2	6.994	Unknown001
3	7.086	Unknown002
4	7.595	Alanine1
5	8.813	Isoleucine1
6	9.191	Valine2
7	11.896	Methionine1
8	12.016	Aspartic acid1
9	12.285	Unknown003
10	13.244	L-proline
11	13.341	Butanoic acid
12	13.378	Glutamic acid1
13	13.504	Unknown004
14	13.613	Phenylalanine1
15	13.797	Unknown005
16	13.994	Unknown006
17	14.398	Glutamic acid2
18	14.494	Phenylalanine2
19	14.819	Unknown007
-	15.587	Ribitol(IS)
20	16.126	Phosphoric acid
21	16.835	Unknown008
22	17.029	Fructose1
23	17.142	Unknown009
24	17.41	Tyrosine
25	17.595	Unknown010
26	18.2	Gluconic acid
27	18.527	Unknown011
28	19.364	M-inosito1
29	20.277	Tryptophan2
30	20.455	Unknown012
31	20.672	Unknown013
32	20.779	Unknown014

Identification of peaks obtained with GC-FID. Numbers indicate the order of substances. Retention times indicate the time the substance appears during the GC-FID. Unidentified peaks were termed unknown.

**Table 2 ijms-23-05300-t002:** Substances with VIP scores greater than 1.0 in the PLS-DA model.

Peak Name	VIP Score	Coefficient
Citric acid	3.18	0.227 **
Threonine2	1.82	−0.130
Lysine1	1.69	0.121
Unknown3	1.60	0.114
Unknown1	1.51	0.108
Lactic acid2	1.27	0.091
Valine2	1.17	0.083
Proline, isolecine1	1.11	0.079

VIP scores were computed using the same data that were used to build the partial least squares-discriminant analysis (PLS-DA) model. The table includes components with a VIP score > 1.0. A No. with multiple component names represents components with peaks that were observed at the same retention time. Peaks whose chemicals could not be identified were labeled ‘Unknown’. One-sided unpaired Students *t*-test; *n* = 4, ** *p* < 0.01.

**Table 3 ijms-23-05300-t003:** IC_50_ of bacteria towards deoxycholic acid with or without glucosylceramide addition.

IC_50_ (mM)			
Bacteria	Control	Glucosylceramide	*p* Value
*L.delbrueckii*	1.17 ± 0.16	1.47 ± 0.07	0.078
*S. epidermidis*	0.343 ± 0.01	0.337 ± 0.00	0.131
*S. mutans*	0.175 ± 0.002	0.251 ± 0.014	0.003 **
*C. striatum*	0.369 ± 0.014	0.393 ± 0.020	0.197
*C. butyricum*	0.088 ± 0.003	0.117 ± 0.014	0.053
*E. coli*	-	-	-
*B. uniformis*	0.21 ± 0.02	0.22 ± 0.02	0.344
*B. coccoides*	0.67 ±0.07	0.81 ± 0.06	0.086
*E. faecalis*	0.06 ± 0.00	0.08 ± 0.00	0.007 **

IC_50_ values were calculated independently for each bacterial culture. The IC_50_ values of the glucosylceramide group are included for comparison with that of the control group. Data are expressed as means ± standard errors. One-sided unpaired Student’s *t*-test; *n* ≥ 3; ** *p* < 0.01. IC_50_ of *E. coli* was not described because its growth increased with deoxycholic acid and it was difficult to calculate IC_50_.

## Data Availability

Inquiries can be directed to the corresponding author.

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
