# Peer review of "Glucosylceramide Changes Bacterial Metabolism and Increases Gram-Positive Bacteria through Tolerance to Secondary Bile Acids In Vitro†"

_ijms, 2022, doi:10.3390/ijms23105300_

Round 1

Reviewer 1 Report

The manuscript submitted by Dai et al., entitled as “Glucosylceramide changes bacterial metabolism and increases Gram-positive bacteria through tolerance to secondary bile acids in vitro” is very interesting and well designed. The experiential are well organized and performed. However, it needs few more experiments to increase the quality of the manuscript, which are as follows:

  1. How authors have decided glucosylceramide concentration 20 ug/mL.
  2. Please provide the primer sequence used for the PCR analysis.
  3. I also recommend evaluating the tolerance of bacteria against bile salt upon glucosylceramide treatment.
  4. Check if the adhesion ability of intestinal microbes on mammals intestinal cells increased after glucosylceramide treatment.
  5. The following article are suitable to cite:

Dey et al., LWT. 2019, 111: 663-672.

Patterson et al., FASEB J. 2020. 34: 15922–15945.

Author Response

Dear respected reviewer,

We really appreciate your precious review of our manuscript.

The manuscript submitted by Dai et al., entitled as “Glucosylceramide changes bacterial metabolism and increases Gram-positive bacteria through tolerance to secondary bile acids in vitro” is very interesting and well designed. The experiential are well organized and performed. However, it needs few more experiments to increase the quality of the manuscript, which are as follows:

>How authors have decided glucosylceramide concentration 20 ug/mL.

This was decided based on the concentration of glucosylceramide in crops and fermented foods. This was added to text.

>Please provide the primer sequence used for the PCR analysis.

We added the citation including primer information to text.

>I also recommend evaluating the tolerance of bacteria against bile salt upon glucosylceramide treatment.

We investigated this, but the effect of bile acids was only marginal and could not determine IC50.

>Check if the adhesion ability of intestinal microbes on mammals intestinal cells increased after glucosylceramide treatment.

We appreciate your important and excellent advice. Since it is beyond the theme of this study, we would like to add this experiment in the next research.

>The following article are suitable to cite:

Dey et al., LWT. 2019, 111: 663-672.

Patterson et al., FASEB J. 2020. 34: 15922–15945.

These were cited.

Reviewer 2 Report

The manuscript entitled “Glucosylceramide changes bacterial metabolism and increases Gram-positive bacteria through tolerance to secondary bile aids in vitro” describes that demonstrate that the addition of glucosylceramide induces changes in the metabolism of B. coccoides and increases the number of Gram-positive bacteria in an intestine-simulating environment through tolerance to deoxycholic acid.

1.In the figure 3, the increase of OD is not notable. The increase of OD600 may be caused by glucosylceramide-induced growth of Gram-positive bacteria instead of the tolerance to secondary bile acids in vitro.

2.Authors should provide more information about Blautia and Gram-positive bacteria.

3.What is the potential for clinical application from the finding of this manuscript?

Author Response

Dear respected reviewer,

We really appreciate your precious review of our manuscript.

The manuscript entitled “Glucosylceramide changes bacterial metabolism and increases Gram-positive bacteria through tolerance to secondary bile aids in vitro” describes that demonstrate that the addition of glucosylceramide induces changes in the metabolism of B. coccoides and increases the number of Gram-positive bacteria in an intestine-simulating environment through tolerance to deoxycholic acid.

>In the figure 3, the increase of OD is not notable. The increase of OD600 may be caused by glucosylceramide-induced growth of Gram-positive bacteria instead of the tolerance to secondary bile acids in vitro.

We apologize for our unclearness. This information should be interpreted from the OD600 values at 0 ug/ml glucosylceramide.

>2.Authors should provide more information about Blautia and Gram-positive bacteria.

This was added to discussion.

>3.What is the potential for clinical application from the finding of this manuscript?

This was added to discussion.

Reviewer 3 Report

The authors need to add to the discussion to present the limitation and weakness of these studies. Specifically adding compounds to monocultures is not likely to reflect what occurs in the complex microbiota of the gut.

It would add physiological relevance if the authors would related the concentrations of glucosylceramide added to the in vitro tests with the amounts that could be feasibly consumed by humans with the amounts shown in references 23-25. Do they match? 

The cell mock 1 group needs to be explained more clearly. I cannot tell whether all 15 are being grown together. It seems they are not since individual medium are being specified for difference bugs. 

Author Response

Dear respected reviewer,

We really appreciate your precious review of our manuscript.

>The authors need to add to the discussion to present the limitation and weakness of these studies. Specifically adding compounds to monocultures is not likely to reflect what occurs in the complex microbiota of the gut.

This was added to discussion.

>It would add physiological relevance if the authors would related the concentrations of glucosylceramide added to the in vitro tests with the amounts that could be feasibly consumed by humans with the amounts shown in references 23-25. Do they match? 

This was added to discussion.

>The cell mock 1 group needs to be explained more clearly. I cannot tell whether all 15 are being grown together. It seems they are not since individual medium are being specified for difference bugs. 

This was added to results.

Round 2

Reviewer 1 Report

I think authors have addressed all the queries point to point. They have done adequate changes which makes it eligible for acceptance.

Reviewer 2 Report

Authors have well revised the manuscript according to reviewers' suggestions.